# Biosynthesis and Biological Significances of LacdiNAc Group on *N*- and *O*-Glycans in Human Cancer Cells

**DOI:** 10.3390/biom12020195

**Published:** 2022-01-24

**Authors:** Kiyoko Hirano, Kiyoshi Furukawa

**Affiliations:** 1Glycoinformatics Project, The Noguchi Institute, Itabashi, Tokyo 173-0003, Japan; 2Department of Endocrinology and Diabetes, Medical Center, Saitama Medical University, Kawagoe 350-8550, Japan; kiyoshif@saitama-med.ac.jp

**Keywords:** LacdiNAc, β4-*N*-acetylgalactosaminyltransferase, tumor malignancy, epithelial-mesenchymal or mesenchymal-epithelial transition, gene therapy

## Abstract

An increasing number of studies have shown that the disaccharide GalNAcβ1→4GlcNAc (LacdiNAc) group bound to *N*- and *O*-glycans in glycoproteins is expressed in a variety of mammalian cells. Biosynthesis of the LacdiNAc group was well studied, and two β4-*N*-acetylgalactosaminyltransferases, β4GalNAcT3 and β4GalNAcT4, have been shown to transfer *N*-acetylgalactosamine (GalNAc) to *N*-acetylglucosamine (GlcNAc) of *N*- and *O*-glycans in a β-1,4-linkage. The LacdiNAc group is often sialylated, sulfated, and/or fucosylated, and the LacdiNAc group, with or without these modifications, is recognized by receptors and lectins and is thus involved in the regulation of several biological phenomena, such as cell differentiation. The occurrences of the LacdiNAc group and the β4GalNAcTs appear to be tissue specific and are closely associated with the tumor progression or regression, indicating that they will be potent diagnostic markers of particular cancers, such as prostate cancer. It has been demonstrated that the expression of the LacdiNAc group on *N*-glycans of cell surface glycoproteins including β1-integrin is involved in the modulation of their protein functions, thus affecting cellular invasion and other malignant properties of cancer cells. The biological roles of the LacdiNAc group in cancer cells have not been fully understood. However, the re-expression of the LacdiNAc group on *N*-glycans, which is lost in breast cancer cells by transfection of the β4GalNAcT4 gene, brings about the partial restoration of normal properties and subsequent suppression of malignant phenotypes of the cells. Therefore, elucidation of the biological roles of the LacdiNAc group in glycoproteins will lead to the suppression of breast cancers.

## 1. Introduction

Alterations of glycan structures of cell surface glycoproteins and glycolipids affect many biological phenomena, such as cell differentiation, immune response, cell adhesion, and the malignant transformation of cells [1]. The GalNAcβ1→4GlcNAc (LacdiNAc) group is widely expressed on *N*- and *O*-glycans in invertebrates, in particular, parasitic helminths [2], but was hardly detected in mammalian glycoproteins in early studies. However, advanced analytical methods for glycan structures have enabled us to show that the LacdiNAc group is also distributed in the glycoproteins of a variety of mammalian cells and tissues. It was also found that this disaccharide group is involved in several biological phenomena, such as the control of the half-life of glycohormones in circulation [3,4]. Here, in this article, we will summarize the biosynthetic pathways and biological significances of the LacdiNAc group of glycoproteins in mammalian cells and discuss how disaccharide is important for human cancers, particularly breast cancer.

## 2. Biosynthesis of LacdiNAc Group and Its Modification in Mammalian Cells

The biosynthetic pathway of the LacdiNAc group is summarized in Figure 1. Two β4-*N*-acetylgalactosaminyltransferases (β4GalNAcTs), β4GalNAcT3 (gene ID: 283358) [5] and β4GalNAcT4 (gene ID: 338707) [6], which are involved in this biosynthesis, are type-II transmembrane proteins and have 43% sequence homology. These glycosyltransferases belong to the human β4-galactosyltansferase family and show high homology to chondroitin sulfate synthase 1 [5,6]. The enzymes transfer *N*-acetylgalactosamine (GalNAc) from UDP-GalNAc to non-reducing terminal *N*-acetylglucosamine (GlcNAc) of both *N*- and *O*-glycans in a β-1,4-linkage with a similar substrate specificity [7,8]. However, the tissue distribution of these is quite different [5,6]. Since β4GalNAcT3 and β4GalNAcT4 have been shown to localize in the trans-Golgi network [9,10] and share the same acceptors with β1,4-galactosyltrasferase I, which is abundantly expressed in most mammalian cells and forms the Galβ1→4GlcNAc (LacNAc) group, this might be the reason why the LacdiNAc group is a minor disaccharide compared to the LacNAc group on *N*-glycans in mammalian cells. It is also noted that some other glycosyltransferases are able to catalyze the formation of the LacdiNAc group. In fact, the overexpression of the *Caenorhabditis elegance* β4GalNAcT gene in Chinese hamster ovarian Lec8 cells resulted in the formation of the poly-LacdiNAc structure [11].

The LacdiNAc group is often modified by sialylation, sulfation, and/or fucosylation. The LacdiNAc group, carrying α-2,6-linked sialic acid, is often detected in mammalian cells [10,12,13,14]. On the other hand, the sulfated form of the LacdiNAc group, which was first detected in the bovine pituitary hormone lutropin [15], is now found in several other human glycoproteins [16,17,18]. Two *N*-acetylgalactosamine-4-*O*-sulfotransferases (GalNAc4STs), GalNAc4ST-1 or CHST8 (gene ID: 64377) [19] and GalNAc4ST-2 or CHST9 (gene ID: 83539) [20], can transfer a sulfate group to the 4-position of the GalNAc residue, resulting in the formation of the sulfated LacdiNAc group [8]. Likewise, the fucosylated LacdiNAc group is detectable in mammalian glycoproteins [13,14,21], and the overexpression of α-1,3-fucosyltransferase in Chinese hamster ovarian Lec8 cells brought about the formation of the fucosylated LacdiNAc group [11]. The modification of the non-reducing terminal glycan groups of many glycoproteins affects their biological roles. For instance, the terminal sialic acid residues are recognized by siglecs, which are sialic acid-binding immunoglobulin-like receptors, and these bindings have been shown to regulate the binding/recognition of immunocytes to target cells [22]. It is not clear, however, whether the modification of the LacdiNAc group affects the function of its carrier proteins; several studies have shown that the modified LacdiNAc group is recognized by the respective receptors and lectins and thus participates in the activation or inhibition of their underlying signal pathways, as will be discussed below.

## 3. Biological Roles of LacdiNAc Group on *N*- and *O*-glycans

Some receptors and lectins recognize the LacdiNAc group on *N*- and *O*-glycans, and these bindings are involved in keeping homeostasis of mammals. The mannose/GalNAc-4-SO_4_ receptor on hepatic endothelial cells binds to the sulfated LacdiNAc group on *N*-glycans carried by a pituitary glycohormone, lutropin, thus resulting in the clearance of this hormone from the blood circulation [16,23]. Likewise, the sialylated LacdiNAc group on *N*-glycans is recognized by the asialoglycoprotein receptor in the liver, and this binding leads to the rapid clearance of the present glycoprotein [24]. It has been reported that a coagulation factor VII-albumin fusion protein, expressed in HEK293 cells, possesses the LacdiNAc group on its *N*-glycans [25]. However, when the β4GalNAcT3 and β4GalNAcT4 genes were knocked out in order to eliminate the GalNAc residue from the LacdiNAc group, the fusion protein failed to bind both to a mannose/GalNAc-4-SO_4_ receptor and an asialoglycoprotein receptor, causing its prolonged half-life in circulation [26]. Because remodeling of the glycan structures of biopharmaceutical molecules, such as immunoglobulins, can provide more effective molecules than the natural ones [27,28], it may be worth mimicking the LacdiNAc group on *N*-glycans of glycoproteins.

Since the LacdiNAc group, expressed abundantly on *O*-glycans of MUC5AC mucin in human gastric mucosa, has been shown to play an important role in the interaction with *Helicobacter pylori* [29,30], specific adhesion molecules to this disaccharide could be present on the bacterium and/or gastric epithelial cells, which has to be pursued in future.

A previous study has shown that galectin-3, which is a mammalian β-galactoside-binding lectin, can bind to the LacdiNAc group on *N*-glycan of macrophage and regulate the immune response [31]. Indeed, in vitro studies have demonstrated that galectin-3 possesses a high binding affinity to the LacdiNAc group through its carbohydrate-recognition domain [32,33]. Quite recently, Sedlář and co-workers have shown that the LacdiNAc disaccharide, added in cell culture medium, inhibits binding of galectin-3 to human adipose tissue-derived stem cells and human umbilical vascular endothelial cells in a concentration-dependent manner [34]. Since galectin-3 is involved in cell-to-substratum interaction [35,36], the glycomimetics of the LacdiNAc disaccharide are now under consideration as a possible therapeutic reagent for several diseases, including cancers [37].

Our previous study has shown that the expression of the LacdiNAc group is closely associated with the functional differentiation of the bovine mammary gland [38]. Furthermore, the LacdiNAc group found on *N*-glycans of the leukemia inhibitory factor receptor of mouse stem cells has been shown to be involved in their self-renewal [39]. Thus, the LacdiNAc group on *N*- and *O*-glycans is quite important for the growth and differentiation of mammalian cells.

## 4. Differential Expression of LacdiNAc Group among Cancer Cells

Due to the limited expression of the LacdiNAc group on human glycoproteins, the precise biological roles of this disaccharide in mammals, especially in human diseases, remain to be elucidated. However, the glycomics and glycotranscriptomics of a variety of human cancer tissues and cell lines have shown that the expression patterns of the LacdiNAc group on *N*- and *O*-glycans and the β4GalNAcTs are different (reviewed in [4]). For instance, increased expression of the LacdiNAc group is observed in glycoproteins of prostate cancer, colon cancer, pancreatic cancer, and ovarian cancer [40,41,42,43,44,45,46,47,48], while decreased expression of this disaccharide is observed in those of neuroblastoma, breast cancer, and gastric cancer [49,50,51,52] (Table 1).

The altered expression of the LacdiNAc group on cell surface and/or secretory glycoproteins is assumed to be a useful biomarker for different types of cancers and even different stages of a certain cancer. One of the benefits of this glycan-based biomarker is a prostate-specific antigen (PSA), which has been used widely as a diagnostic marker of prostate cancer. The serum PSA level of patients with prostate cancer increases compared to those of non-cancer counterparts; however, it is still difficult to discriminate between patients with prostate cancer and those with benign prostate hyperplasia (BPH) when the concentration of PSA is low in sera, which is called a gray zone. Several groups have reported that the amount of PSA carrying *N*-glycans with the sialylated LacdiNAc group increases significantly in the patient’s sera compared to those of BPH [40,41,42]. These findings opened several new diagnostic approaches, including a highly sensitive mass-spectrometry-based analysis of glycan structures of PSA and a sandwich enzyme-linked immunosorbent assay system coupled with a PSA antibody and *Wisteria floribunda* agglutinin (WFA), which recognizes a terminal GalNAc residue, to discriminate patients with prostate cancer in a gray zone from those with BPH [42,43,44]. It should be noted that the expression level of β4GalNAcT4 gene is up-regulated compared to that of β4GalNAcT3 gene in prostate cancer, suggesting that β4GalNAcT4 is responsible for the formation of the LacdiNAc group in prostate cancer [40]. Likewise, the enhanced expression of the LacdiNAc group is observed in a particular type of ovarian cancer cells such as serous adenocarcinoma and clear cell carcinoma [17,47,48]. In contrast, when the tissues from patients with breast cancer were analyzed histochemically by staining with WFA, strong staining was observed in the non-malignant regions of the specimens and decreased or non-staining was observed in the progressed regions of this cancer [50], indicating that the LacdiNAc group on *N*-glycans is important for the maintenance of normal functions of mammary epithelial cells. Taken together, these findings strongly indicate that the detection of different expression levels of the LacdiNAc group and/or certain modified LacdiNAc forms might be a useful and sensitive diagnostic and/or prognostic marker of the respective cancers.

## 5. Occurrence of LacdiNAc Group on *N*-glycans of Cell Surface Molecules

Several studies have shown that the expression of the LacdiNAc group on *N*-glycans of cell surface molecules such as β1-integrin seems to be closely associated with tumor progression or regression, depending on the types of cancer cells, as shown in Table 2.

Over-expression of the β4GalNAcT3 gene in HCT116 colon cancer cells induced the production of the LacdiNAc group on *N*-glycans of β1-integrin and resulted in the promotion of migratory and invasive activities of the cells, and of the adhesive activities of the cells toward fibronectin, collagen-type IV, and laminin, respectively [45]. When the β4GalNAcT3 gene was knocked down in these colon cancer cells, the decreased expression of the LacdiNAc group on *N*-glycans of the epidermal growth factor receptor (EGFR) resulted in the inhibition of the phosphorylation of EGFR and its downstream signaling molecules, AKT and ERK, followed by the degradation of EGFR [54]. Furthermore, this gene knock down was accompanied with a decrease in sphere formation of the cells and the expression of stem cell markers such as OCT4, indicating that the LacdiNAc group on *N*-glycans is important for maintaining the stemness of the cancer cells [54].

In contrast, enhanced expression of the LacdiNAc group on cell surface *N*-glycans, including those of β1-integrin by the transfection of the β4GalNAcT3 gene into SK-N-SH and SH-SY5Y neuroblastoma cells, showed inactivation of signal transduction pathways mediated with FAK, AKT, and ERK and decreased binding to extra-cellular matrices, particularly to laminin [49]. Therefore, the expression of the LacdiNAc group on *N*-glycans of β1-integrin appears to regulate malignant properties positively or negatively, depending on the types of cancer cells.

We have shown that the enhanced expression of the β4GalNAcT4 gene in MDA-MB-231 human breast cancer cells in order to re-express the LacdiNAc group on cell surface glycoproteins, as are expressed, presumably, in normal human mammary gland epithelial cells [50], suppressed colony formation, in vivo tumor formation, and in vitro invasion [10]. Quite interestingly, MDA-MB-231 cells re-expressed the LacdiNAc group on their *N*-glycans, which were well spread and enlarged, and expressed an epithelial cell marker, E-cadherin, highly at the cell surface [53], indicating the induction of mesenchymal–epithelial transition (MET). Moreover, MDA-MB-231 cells expressing the LacdiNAc group on their *N*-glycans showed stronger adhesion toward fibronectin, collagen-type I, collagen-type IV, and laminin when compared to those of the control cells. There are several reports describing that the changing of the glycosylation patterns of cell surface glycoproteins artificially can induce MET as well as epithelial-mesenchymal transition (EMT) in human breast cancer cells, depending on the conditions of the glycosylation. For instance, increased expression of α2,6-linked sialic acid residues by transfection of the α2,6-sialyltransferase I gene induced EMT in MDA-MB-231 cells [55]. Thus, our finding indicates that the re-expression of the LacdiNAc group on *N*-glycans may be involved in the induction of a MET-like change in the β4GalNAcT4 gene-transfected breast cancer cells. Although it is unclear how the LacdiNAc group on *N*-glycans induces a MET-like change in the breast cancer cells, our preliminary studies showed that the expression of the LacdiNAc group occurs predominantly on *N*-glycans of β1-integrin in the gene-transfected cells (unpublished data), and that β1-integrin carrying *N*-glycans with the LacdiNAc group may play some crucial role in this induction through its binding to a selected α-subunit species and/or a particular extracellular matrix. Integrins are heterodimeric cell surface receptors composed of α- and β-subunits and bind selectively to extracellular proteins such as fibronectin and laminin [56]. Both subunits possess several potential N-glycosylation sites, and the glycosylation of each subunit is essential, not only for formation of heterodimers but also for certain biological roles such as cell-to-cell and cell-to-substratum interactions [57,58,59]. For instance, α5β1-integrin-mediated cell spreading is inhibited when *N*-glycans of the integrin are modified with a bisecting GlcNAc residue upon over-expression of the β4-*N*-acetylglucosaminyltransferase III gene in Hela S3 cells and others [58,60,61]. Thus, to understand the biological roles of the LacdiNAc group on *N*-glycans in the MET-induced breast cancer cells, it is important to study which N-glycosylation site(s) of β1-integrin are modified with the LacdiNAc group and whether this glycan affects the dimer formation with a particular α-subunit or not.

It has been reported that interaction between galectin-3 and integrins significantly affects an adhesive property to extracellular matrices, such as laminin, in a variety of cells [34,35]. However, the effect of galectin-3 on malignant properties of cells is different in each tumor, and galectin-3 promotes adhesion to laminin in several breast cancer cell lines [62,63]. Our preliminary data showed that galectin-3 binds to the LacdiNAc group in the β4GalNAcT4 gene-transfected breast cancer cells (unpublished data), indicating that the possible interaction of galectin-3 with the LacdiNAc group on *N*-glycans of integrins may play a role in the regulation of cell-to-substratum interaction, thus contributing to the induction of the MET. 

As described above, the expression of the LacdiNAc group on *N*-glycans of β1-integrin exhibits opposite effects on the regulation of malignant properties among cancer cell types. It should be noted that the *N*-glycans with the LacdiNAc group may occur not only in β1-integrin but also in many other glycoproteins, including EGFR, which causes more complicated regulation of tumor malignancy (Figure 2). Detailed studies are required to find out biological roles of the LacdiNAc group on *N*-glycans in individual glycoproteins.

The enhanced expression of a glycosyltransferase gene whose expression is lost by malignant transformation of cells by a gene-transfer technique appears to be a promising approach to suppress the growth of tumors in vivo [64,65,66]. In fact, the introduction of the β4GalNAcT4 gene into MDA-MB-231 human breast cancer cells to express the LacdiNAc group on the *N*-glycans again [50] resulted in the loss of an ability that can form tumors in athymic mice [10].

## 6. Conclusions

The LacdiNAc group, with or without modification, plays important roles in many biological phenomena, especially tumor progression and regression. The monitoring of the expression levels of this disaccharide group may be useful as a diagnostic/prognostic marker of particular tumors. The detailed controlling mechanism of the malignant properties of cancer cells by this glycosylation remains to be elucidated, but the β4GalNAcT3 gene, the β4GalNAcT4 gene, and their antisense oligonucleotides have to be evaluated as new cancer therapy agents for several tumors.

## Figures and Tables

**Figure 1 biomolecules-12-00195-f001:**
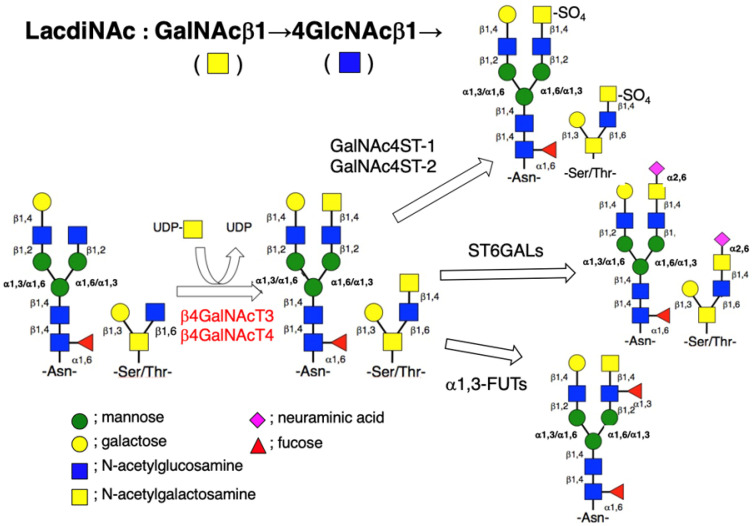
Biosynthetic pathway of *N*- (Asn-linked) and *O*- (Ser/Thr-linked) glycans with the LacdiNAc group in mammalian cells. The β4GalNAcT3 and β4GalNAcT4 can transfer GalNAc from UDP-GalNAc to the non-reducing terminal GlcNAc residue of both *N*- and *O*-glycans in a β-1,4-linkage to produce the LacdiNAc group. In some cases, this disaccharide receives sialylation, sulfation, and/or fucosylation.

**Figure 2 biomolecules-12-00195-f002:**
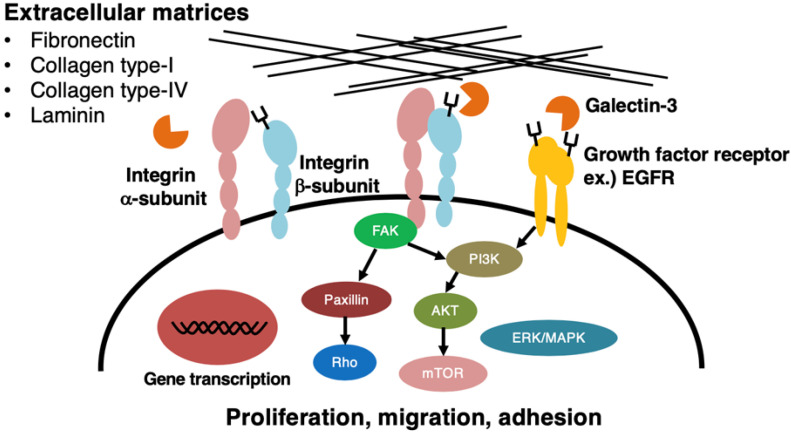
Activation or inactivation of signal transducing molecules, followed by modulation of malignant properties of cancer cells upon expression of LacdiNAc group on *N*-glycans of cell surface molecules. Changes in the expression level of the LacdiNAc group on *N*-glycans of β1-integrin and/or EGFR affect signal transducing molecules, including FAK and AKT, and most probably, Paxillin and Rho result in changes of proliferation, migration, and/or adhesion of cancer cells. In some cases, the LacdiNAc group on *N*-glycans of these cell surface molecules may interact with galectin-3 and affect cell-to-substratum interaction.

**Table 1 biomolecules-12-00195-t001:** Differential expression of LacdiNAc group on *N*-glycans among human cancer tissues and cells.

Types of Tumor/Cancer Cells	References
**Positive correlation**	
Prostate cancer cells	[40,41,42,43,44]
Colon cancer cells	[45]
Pancreatic cancer cells	[46]
Ovarian cancer tissues and cells	[47,48]
**Negative correlation**	
Neuroblastoma tissues and cells	[49]
Breast cancer tissues and cells	[50]
Gastric cancer tissues and cells	[51,52]

**Table 2 biomolecules-12-00195-t002:** Differential effects of LacdiNAc group expressed in *N*-glycans on malignant properties of human cancer cells.

Malignant Properties	References
**Proliferation**	
promoted in HCT116 colon cancer cells	[45]
suppressed in SK-N-SH/SH-SY5Y neuroblastoma cells	[49]
suppressed in MDA-MB-231 breast cancer cells	[10]
**Migration and invasion**	
promoted in HCT116 colon cancer cells	[45]
suppressed in SK-N-SH/SH-SY5Y neuroblastoma cells	[49]
suppressed in MDA-MB-231 breast cancer cells	[10]
**Adhesion to extracellular matrices**	
promoted in HCT116 colon cancer cells	[45]
suppressed in SK-N-SH/SH-SY5Y neuroblastoma cells	[49]
promoted in MDA-MB-231 breast cancer cells	[53]
**Mesenchymal-epithelial transition**	
induced in MDA-MB-231 breast cancer cells	[53]

## Data Availability

Not applicable.

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
