# Peer review of "Biosynthesis and Biological Significances of LacdiNAc Group on N- and O-Glycans in Human Cancer Cells"

_biomolecules, 2022, doi:10.3390/biom12020195_

Round 1

Reviewer 1 Report

The manuscript by Hirano and Furukawa provides an important review of the current state of knowledge regarding the biological significance of the LacdiNAc group in cancer cells. The authors summarize the relevant literature and then synthesize this information into a coherent evaluation of the biological roles of the LacdiNAc group on N- and O-glycans, especially in the context of cancer.

Minor points

1) In the abstract, I consider the following statement to be premature "the introduction of the respective glycosyltransferase gene into tumors will be encouraged in future cancer therapy" (lines 26-27). More studies are needed to know whether this type of strategy will be encouraged in the future.

2) Figure 1: Regarding the statement "The terminal GalNAc residue of the LacdiNAc group often receives further modification such as sialylation, sulfation, and/or fucosylation" (lines 60-61), it is necessary to clarify some points:

     a) This statement, as written, suggests that fucosylation would occur in GalNAc, however, this does not seem to be possible.

     b) Is it possible to have this type of fucosylation in O-glycans? Has this already been described? In this case, it is recommended to include this possibility in Figure 1.

     c) It is possible to find the LacdiNAc group carrying a2,3 sialic acid in mammalian cells? Or is it just a2,6?

3) Line 108: glycomimetrics or glycomimetics?

4) Line 175 (Reference 50): Here it would be interesting to add that the knockdown of B4GALNT3 suppresses the expression of stem-cell associated markers (OCT4 and NANOG), which suggests that LacdiNAc termini on N-glycans contribute to the maintenance of cell stemness.

Reviewer 2 Report

In the research paper entitled “Biosynthesis and biological significances of LacdiNAc group on N- and O-glycans in human cancer cells”, the authors tried to give brief information about the significance of LacdiNAc marks on glycans. There are a few drawbacks authors should consider

  1. The information quoted in the paper is quite old (only 12 references out of 65 were published after 2015)
  2. As there is already a review published in 2014(PMID: 25003135) by the same group, this review should be like a recent update on developments in the field of LacdiNAc group on N- and O-glycans.

Reviewer 3 Report

The review from hirano and Furukwa describes the biosynthesis and biological significances of a very specific glycosylation process, the presence of LacdiNac group on N- and O-glycans in human cancer cells. This is an interesting review, well written and detailed but some parts are floppy. Please find my suggestions to tentatively improve them.

I personal found the paragraph on the biosynthesis quite minimalist. Although I recognize the importance of describing the enzymes involved in such biosynthesis, the authors could have mentioned what is known on the regulation of such biosynthesis, mention the issue of the competition with the galactosylation, the place/ expression levels of the enzymes in the different cisternae of the Golgi for example. The difference between type I and Type II is not evoked and at the end we don’t know in the rest of the paper of which type we are talking. It seems important to mention to me. 

Even if some of these questions are unanswered yet, I have the feeling that it would give to the readers a better view of the complexity of such biosynthesis. Nothing is mentioned on the possibility to have poly LacdiNac. Although Galβ1–4GlcNAc (LacNAc) moieties are the most common constituents of N-linked glycans on vertebrate proteins, GalNAcβ1–4GlcNAc containing glycans are widespread in invertebrates, such as helminths. This should also be mentioned somewhere especially since the next paragraph deals with the biological roles. 

The part on the differential expression is interesting but asks questions. We don’t know whether the studies have shown an increased biosynthesis of these groups or the lack of terminal sialylation that “unmask” these structures?? I ask this since later in the paragraph the authors mention the use of WFA that recognizes unmasked structures. Would the phenotype due to the increased presence of LacdiNAc structures of the lack if sialylation of these structures. For non-specialists, I believe this is important to be precise on that part. 

Paragraph 5: the part on Beta integrin is clearly interesting. Nothing is known on the stability of beta 1 integrin and its glycosylation status?? I suppose that the involvement of beta 1 integrin on the described biological processes depends on its quantity.

Concerning the paragraph 6, I’m not sure to follow the logic of the authors and as it is it seems a bit out of the scope of the review. Should better be incorporated in paragraph 4 no?

Maybe a paragraph on the available tools to study such groups could be informative.

Round 2

Reviewer 2 Report

Th authors answered all the questions raised

Reviewer 3 Report

none